

# Exploring the impact of curiosity and sport commitment on creativity among fitness coaches: the mediating role of knowledge-sharing and flow experience

Ying Chen[1] and Bin Chen[2]

[1] School of Law, Fuzhou University, Fuzhou, Fujian, China
[2] Department of Public Physical Education, Fujian Agriculture and Forestry University, Fuzhou, Fujian, China

## ABSTRACT

The creativity of fitness coaches is crucial for maintaining the competitiveness of fitness enterprises. This study aimed to explore the factors, mechanisms, and moderating effects influencing the creativity of fitness coaches ($n = 732$) in Chinese fitness businesses, using partial least squares structural equation modeling (PLS-SEM) analysis. The results indicate that curiosity and sport commitment have a positive impact on knowledge-sharing and flow experience. In addition, knowledge-sharing and flow experience positively influence creativity. Importantly, the curiosity and sport commitment of fitness coaches affect creativity through knowledge-sharing and flow experience. Finally, social media usage positively moderates the relationship between knowledge-sharing and creativity. The findings of this study provide meaningful information and decision-making references for professionals in the fitness industry and academic researchers. Future efforts should emphasize the protection of fitness coaches' creative achievements, with policymakers encouraged to establish relevant legal frameworks to safeguard innovation within the industry.

## INTRODUCTION

Early 2020 saw the global fitness business being impacted by the COVID-19 pandemic. According to a report by *GWI (2021)*, the fitness market revenue in 2020 was $77.2 billion, a 37.1% decrease compared to $122.8 billion in 2019. However, the number of paid gym memberships in China is on the rise. For instance, in 2021, the number of paid fitness members reached 75.13 million, showing a 6.89% increase compared to 2020. Although the penetration rate of fitness among the overall population in China is only 5.37%, China is one of the countries where the number of fitness members is growing (*CFIDR, 2021*). The growth can be attributed to China's economic restructuring, consumer upgrading, and the deepening of health awareness. Despite the end of the pandemic, the fitness business in China is still facing many challenges, including declining numbers of businesses, losing

Corresponding author
Bin Chen, chenbin@fafu.edu.cn

coaches, and fierce price competition. Therefore, Chinese fitness enterprises need to formulate a healthy, sustainable, and effective marketing strategy.

In recent years, Chinese fitness enterprises have been facing a process of survival of the fittest and intense competition. According to a report by *CFIDR (2022)*, the fitness industry experienced a positive growth rate of 4.0% compared to negative growth in 2020. Many fitness companies are undergoing digital transformation by leveraging technology, such as internet fitness models. Currently, although online fitness models cater to diverse consumer demands, two distinctly different outcomes have emerged. Some fitness enterprises have successfully transitioned, such as the "Liu Genghong Phenomenon" which has driven the online fitness craze through platforms like Douyin (*CFIDR, 2022*). However, some fitness enterprises have failed in their transition, possibly due to issues like mismatched service quality, poor communication between coaches and members, and lack of freshness and effectiveness in courses. Therefore, for fitness companies, the creativity of their employees is crucial in enhancing the quality of online fitness course services. More specifically, the creativity of a fitness coach is an important strategic resource for the competitiveness of a fitness enterprise, particularly in the evolving digital fitness landscape where differentiation through unique and engaging training programs is key to success.

Currently, scholars in the field of sports creativity have explored various aspects such as motor creativity (*Richard et al., 2018*), tactical creativity (*Furley & Memmert, 2018*), creativity in team sports (*Santos et al., 2016*), and sport organizational creativity (*Paek et al., 2022*; *Smith & Green, 2020*). These studies primarily focus on competitive sports, team-based strategies, and organizational-level innovation, involving athletes, coaches, organizations, and administrative personnel. However, the fitness coaching industry differs significantly from competitive sports as it focuses on personalized service, client motivation, and the ability to design adaptive and innovative fitness programs (*Eastabrook & Collins, 2020*). On the other hand, research in the fitness domain has explored service quality (*Jeon et al., 2021*), service convenience (*García-Fernández et al., 2018*), satisfaction (*Fernández-Martínez et al., 2020*), and switching costs (*Kim & Byon, 2021*), mainly from the perspectives of consumers or managers. Although there are significant findings in sports creativity and the fitness domain, the creativity of fitness coaches remains largely unexplored, particularly in terms of the personal and contextual factors that foster their creativity. The personal creativity of fitness coaches can provide innovative raw materials for fitness enterprises, and a better understanding of their creativity will enhance the innovation capabilities of these enterprises (*Amabile, 1988*). The creativity of fitness coaches is crucial for enhancing customer experience, developing innovative fitness programs, and maintaining competitiveness in the industry (*Acikgoz, Filieri & Yan, 2023*). Therefore, a deeper understanding and cultivation of the creativity of fitness coaches hold significant value for the sustainable development and innovation of the entire fitness industry.

Research indicates that employee creativity is influenced by both personal and professional traits (*Amabile et al., 1996*; *Oldham & Cummings, 1996*). Although existing research has enhanced our understanding of the factors affecting creativity, such as personality traits (*Li et al., 2017*), self-efficacy (*Puozzo & Audrin, 2021*), social networks (*Luo et al., 2021*), and leadership behavior (*Lee et al., 2020*), the impact of curiosity and

sport commitment on the creativity of fitness coaches remains an unexplored area. Firstly, curiosity as a personal trait has been widely recognized as a significant influencer in stimulating individual creative thinking and innovative behaviors (*Kashdan, Rose & Fincham, 2004*). However, existing studies on curiosity have focused on education (*Kashdan & Yuen, 2007*) and management (*Tsai & Zheng, 2021*), with little attention to service-oriented contexts such as fitness coaching, which demands real-time adaptation and individualized program design. Secondly, sport commitment, a professional trait referring to one's determination to invest time and energy in sports to achieve personal goals (*Scanlan et al., 2016*), has been underexplored in the fitness coaching context, even though it is directly related to coaches' job stability, professional identity, and client satisfaction. While prior studies have examined various forms of commitment, such as psychological (*Cheng, Hung & Chen, 2016*), affective commitment, and normative commitment (*Luo et al., 2021*), these have rarely addressed how sport commitment specifically relates to creativity. Notably, how curiosity and sport commitment influence the creativity of fitness coaches in the fitness domain remains to be further explored.

This study aims to explore how curiosity and sport commitment influence creativity in fitness coaches, filling a gap in existing literature. Given the unclear mechanisms of how curiosity and sport commitment affect creativity, the study introduces knowledge-sharing and flow experience as mediating variables. Although prior studies have examined these mediators (*Cheng, Hung & Chen, 2016*; *Luo et al., 2021*; *Schutte & Malouff, 2020*; *Tsai & Zheng, 2021*), most were conducted in educational or organizational contexts. However, fitness coaching involves intense interpersonal interaction, real-time feedback, and adaptive instruction, which may alter how knowledge-sharing and flow function as creativity mechanisms. Thus, re-examining these mediators in the service-oriented context of fitness coaching can improve our understanding of creativity in sport. Knowledge-sharing is considered an external motivator driving creativity, facilitating the exchange of information and ideas crucial for innovation (*Kim & Lee, 2013*). In fitness coaching, knowledge-sharing is vital for exchanging training strategies and client-centered solutions in real time, setting it apart from general corporate settings. Flow experience, an essential intrinsic motivator (*Csikszentmihalyi, 1990*), represents an individual's complete immersion and focus in an activity. It is particularly relevant for fitness coaches, who must remain responsive and fully engaged during dynamic, client-driven sessions.

Moreover, while knowledge-sharing is generally viewed as beneficial for creativity, recent studies have yielded inconsistent results. *Rudawska (2019)* found that proactive and reactive knowledge-sharing have differing effects, with reactive sharing potentially undermining creativity. These variations suggest that contextual factors may shape the effectiveness of knowledge-sharing on creativity (*Men et al., 2019*). To address this, the present study investigates social media usage as a moderating factor. Notably, in the digital age, social media usage is an important contextual factor influencing creativity (*Zhang & Mao, 2025*). It facilitates the rapid exchange of knowledge and promotes interaction among fitness coaches, potentially strengthening the impact of knowledge-sharing on creativity.

This study makes three key contributions to the literature in the fitness domain. Firstly, it proposes an empirical model that explores how curiosity and sport commitment

influence creativity among fitness coaches, specifically within the rapidly evolving and digitally mediated Chinese fitness coaching environment. This offers a novel perspective by highlighting the distinctive creative challenges faced by fitness professionals in China. Secondly, by examining knowledge-sharing and flow experience as mediators, the study reveals the specific mechanisms through which curiosity and sport commitment impact creativity, expanding the understanding of the influence of knowledge-sharing and flow experience on creativity. Lastly, the study provides new insights into the contextual factors of creativity by investigating the moderating role of social media usage in the impact of knowledge-sharing and flow experience on creativity. Overall, this study not only contributes new theoretical perspectives to the literature in the fitness field but also offers valuable insights for practitioners to foster the study and development of creativity among fitness coaches.

## Curiosity, flow experience, and knowledge-sharing

The first proposal of curiosity as a fundamental personality trait was made by *James (1890)*, and his argument that curiosity contains multiple dimensions has sustained the focus of several scholars (*Kashdan et al., 2020*). In their research, *Kashdan et al. (2018)* defined curiosity as the aspiration to identify, follow, and investigate new, uncertain, intricate, and unclear phenomena in their study. Specifically, deprivation sensitivity is defined as the anxiety and frustration associated with wanting to know information that one currently lacks; joyous exploration refers to the pleasurable experience of discovering interesting aspects of the world; social curiosity involves an interest in obtaining information about others' thoughts, feelings, and behaviors; and thrill-seeking represents the excitement that arises when seeking new experiences. For fitness coaches, curiosity not only signifies their proactive approach to solving training-related problems but also enables them to maintain effective communication with clients, thereby enhancing client satisfaction and promoting company performance growth. This research defines curiosity as the degree to which fitness coaches exhibit thrill-seeking, social curiosity, deprivation sensitivity, and joyous exploration when it comes to new concepts, techniques, and knowledge in fitness instruction and training.

*Ardichvili, Page & Wentling (2003)* were among the first to understand that knowledge-sharing involves both knowledge-collecting and knowledge-donating. Knowledge-collecting refers to consulting colleagues to gain their intellectual capital while the transfer of one's intellectual capital to others is known as knowledge-donating (*Van Den Hooff & De Ridder, 2004*). For fitness coaches, being willing to actively learn and help clients in addressing instructional or training issues, as well as sharing professional knowledge, skills, experiences, and fitness information with colleagues (*Kim & Lee, 2013*), can enhance the organization's capabilities in creativity, innovation, and meeting information needs between consumers and colleagues, thereby improving organizational competitiveness. Thus, in the present study, knowledge-sharing is understood as the behavior of fitness coaches who engage in both knowledge-collecting and knowledge-donating with clients and colleagues.

Existing research indicates that curiosity has a significant positive influence on knowledge-sharing. For example, *Tsai & Zheng (2021)* conducted a study with frontline employees to examine the effect of curiosity on service creativity. The study found that curiosity significantly positively influences knowledge-sharing. Furthermore, *Jiang et al. (2022)* conducted a study on 372 manufacturing employees, discussing the influence of psychological contracts on employees' creative performance and the moderating effect of job curiosity. It found that job curiosity positively moderated the relationship between knowledge-sharing and workers' creative performance. Based on all this, it is possible to predict that curiosity will have a significant positive impact on knowledge-sharing. In the context of this study, it is hypothesized as follows:

**H1:** The curiosity of fitness coaches will positively influence knowledge-sharing.

Since *Csikszentmihalyi (1975)* introduced the flow experience, it has become a popular topic of research. Flow experience is defined as "a holistic sense that emerges when we are fully immersed in an activity" (*Csikszentmihalyi, 1975*). According to flow theory, the optimal, temporary, enjoyable, and if a balance exists between perceived challenges and capabilities, a focused, positive psychological state will result (*Hoffman & Novak, 1996*). In this state, individuals experience long periods of autonomous concentration on an activity, losing track of time and becoming absorbed in the task at hand. However, if there is an imbalance between perceived challenges and skills, individuals may experience apathy, anxiety, and boredom (*Csikszentmihalyi, 1990*). In the context of this study, as fitness coaches face different clients and need to fully concentrate and invest time in designing various fitness programs, flow experience is defined as the holistic sense that fitness coaches experience when they are fully engaged in teaching. This includes unambiguous feedback, clear goals, concentration on task at hand, transformation of time, and autotelic experience, all of which contribute to a positive psychological state.

Research suggests that curiosity significantly positively influences flow experience. For instance, people who are more curious are more likely to have flow experience (*Peterson et al., 2007*). Additionally, *Schutte & Malouff (2020)* conducted a study of Australian university students to examine the impact of curiosity on creativity. The research found a significant relationship between curiosity and higher flow experience. Based on the aforementioned literature, curiosity has a positive impact on flow experience. For this study, it is hypothesized as follows:

**H2:** The curiosity of fitness coaches will positively influence the flow experience.

## Sport commitment, flow experience, and knowledge-sharing

Since the introduction of the commitment model in sports by *Scanlan et al. (1993b)*, research on sport commitment has increased significantly in recent years. It is understood as a professional trait reflected in long-term adherence to a particular sport, including dimensions such as enthusiastic commitment, constrained commitment, and sport enjoyment (*Scanlan et al., 2016*). Enthusiastic commitment represents the determination to remain involved in a sport over the long term; constrained commitment reflects the perception of obligation to maintain involvement in a sport; sport enjoyment refers to the positive affective responses experienced during sports participation, reflecting a general

sense of happiness (*Scanlan et al., 2016*). Current research suggests that sport commitment is a crucial variable for explaining athlete participation in sports. For example, those with low sport commitment are more likely to stop doing it (*Bandura & Kavussanu, 2018*). For fitness coaches, who embody the dual roles of athletes and coaches, their commitment to fitness training and instruction can be better understood by examining sport commitment. Understanding this commitment is valuable for comprehending the reasons behind fitness coaches' perseverance in their fitness work and teaching. Therefore, this study defines sport commitment as the professional trait of fitness coaches long-term adherence to fitness training and instruction.

Empirical studies have shown that different types of commitment (organizational commitment, emotional commitment, *etc.*) have an impact on knowledge-sharing. For example, *Luo et al. (2021)* conducted a study with 751 participants in virtual communities to explore the influence of commitment on knowledge-sharing. The study found that commitment has an impact on the willingness to share knowledge. Furthermore, *Ouakouak & Ouedraogo (2019)* examined 307 employees in Canada to investigate the impact of professional trust, personal trust, and affective commitment on knowledge use. The study found that emotional commitment significantly positively impacts knowledge-sharing. Based on the above research, we can see that commitment would impact knowledge-sharing. In the context of this study, it is hypothesized as follows:

**H3:** The sport commitment of fitness coaches will positively influence knowledge-sharing.

Research has shown that commitment has a positive effect on flow experience. *Cheng, Hung & Chen (2016)* aimed to investigate the influence and mechanism of leisure involvement on flow experience. The study found that psychological commitment plays a key mediating role in how leisure involvement influences flow experience. Additionally, *Rivkin, Diestel & Schmidt (2018)* investigated the beneficial effects of flow experience on affective commitment and the buffering moderating effect of self-control demands on well-being indicators in a study of 90 employees. The results showed that flow experience moderated the strong correlation between affective commitment and daily happiness. Based on the above studies, it can be concluded that commitment would impact the flow experience. This study hypothesizes that:

**H4:** The sport commitment of fitness coaches will positively influence the flow experience.

## Knowledge-sharing, flow experience, and coach creativity

Since *Guilford*'s (*1950*) speech on "creativity" at the American Psychological Association, research on the creativity system has been initiated. However, there is currently no unified standard understanding of creativity among different researchers. For instance, *Amabile (1993)* suggested that creativity is the application of novel and useful ideas by employees. For fitness enterprises, creativity is considered a strategic resource that organizations view as a driving force for gaining competitive advantage, organizational survival, and development (*Ghafoor & Haar, 2022*). Stimulating coaches' creativity is beneficial for companies to achieve good organizational performance and sustainable development (*Zhan, Zhang & Trimi, 2023*). Therefore, in this study, coaches' creativity is defined as the process in which

coaches generate innovative ideas and put them into practice, gaining recognition from others and enhancing the performance of themselves, the team, and the organization.

Research has shown that knowledge-sharing has an impact on creativity. For example, *Tsai & Zheng (2021)* examined frontline employees in companies to investigate the impact of curiosity on creativity. Their research found that knowledge-sharing was associated with service creativity. Additionally, *Tuan (2020)* explores the impact of coaches' humility on players' creativity. The study found that knowledge-sharing was an important reason for the effect of coaching humility on players' creativity. In summary, knowledge-sharing has an impact on creativity. The following hypothesis is made in the context of this study:

**H5:** The knowledge-sharing of fitness coaches will positively influence creativity.

Existing research has shown that flow experience has an impact on creativity. For example, *Schutte & Malouff (2020)* conducted a study of 57 Australian university students to explore the impact of curiosity on creativity. Research has found a significant connection between flow experience and creativity. Furthermore, *Chen et al. (2022a)* conducted a survey of 95 employees in a creative industry park to investigate the effects of mindfulness meditation on creativity, flow, and mood. The study revealed that mindfulness, flow, and positive affect all contribute to enhancing participants' work creativity. In summary, flow experience has an impact on creativity. In the study, it is hypothesized that:

**H6:** The flow experience of fitness coaches will positively influence creativity.

## The mediating effect of knowledge-sharing and flow experience

Research has shown that knowledge-sharing is an important mechanism for the influence of curiosity and commitment on creativity. According to *Tsai & Zheng (2021)*, a study of 822 frontline workers in the service industry found that knowledge-sharing is a key reason why curiosity affects creativity. In addition, the existing literature has confirmed the impact of commitment on knowledge-sharing. For example, *Luo et al. (2021)* collected 751 samples from four popular virtual communities through online surveys to explore the impact of commitment on knowledge-sharing. The study found that commitment has an impact on the willingness to share knowledge. In summary, knowledge-sharing is an important mechanism for the influence of curiosity and commitment on creativity. In the study, it is hypothesized that:

**H7:** The curiosity of fitness coaches will positively influence creativity through the mediating effect of knowledge-sharing.

**H8:** The sport commitment of fitness coaches will positively influence creativity through the mediating effect of knowledge-sharing.

Research has shown that flow experience is the mechanism by which curiosity and commitment influence creativity. Studies conducted by *Schutte & Malouff (2020)* found that curiosity has an impact on creativity through the mediating effect of flow experience, supporting the existence of the mediating effect of flow experience. Additionally, *Cheng, Hung & Chen (2016)* conducted a study with hikers in parks to investigate the influence and mechanism of leisure involvement on flow experience. The study found that psychological commitment is an important reason for the influence of leisure involvement on flow experience. In summary, flow experience is an important reason for the influence of

curiosity and commitment on creativity. Accordingly, the following hypotheses are proposed:

**H9:** The curiosity of fitness coaches will positively influence creativity through the mediating effect of flow experience.

**H10:** The sport commitment of fitness coaches will positively influence creativity through the mediating effect of flow experience.

## The moderating effect of social media usage

The term "social media" was first introduced in 1997 (*Boyd & Ellison, 2007*). *Kaplan & Haenlein (2010)* and *Ali-Hassan, Nevo & Wade (2015)* defined social media as encompassing social use, hedonic use, and cognitive use. Social use refers to using social media platforms to establish new social connections, find like-minded individuals, and maintain existing friendships (*Ali-Hassan & Nevo, 2009*); hedonic use emphasizes the enjoyment derived from using social media (*Nevo & Nevo, 2011*); and cognitive use focuses on content creation, sharing, and accessing user-generated content (*Ali-Hassan & Nevo, 2009*). Consequently, social media enables individuals to create, share, and exchange information anytime and anywhere through platforms like Douyin, WeChat, and others (*Ngai, Tao & Moon, 2015*). For fitness coaches, social media serves various purposes. It allows them to stay connected with clients, enhance client satisfaction and loyalty, and gather customer feedback. Additionally, fitness coaches can utilize social media to understand client needs better and develop new fitness programs. In this study, social media usage by fitness coaches is defined as the creation, sharing, and exchange of work-related content with clients, employees, and supervisors through social media platforms such as Douyin, WeChat, Weibo, and others.

Studies have shown that social media usage has an impact on knowledge-sharing. For example, *Cui et al. (2019)* conducted a study of 600 frontline employees to explore the effects and mechanisms of team social media usage on individual work performance. The research finds that knowledge-sharing is an important reason for the impact of team social media usage on job performance. Furthermore, *Rasheed et al. (2020)* discussed the influence of social media usage on students' participation and creativity and the mediating role of knowledge-sharing. It is found that knowledge-sharing is an important reason for the influence of social media usage on students' participation and creativity.

In addition, studies have shown that the use of social media has an impact on creativity. For example, *Luqman et al. (2021)* used Chinese employees as research subjects to explore the impact of social media usage on creativity and well-being. The study found that enterprise social media usage positively influences employee creativity and well-being through the mediating role of psychological transitions. *Zhou et al. (2022)* took the organizations and employees of 172 Chinese knowledge-intensive companies as research objects to explore the relationship between social media usage and employee creativity. The study found that hedonic, cognitive, and social use of social media had an impact on employee creativity.

In summary, the use of social media has an impact on knowledge-sharing and creativity. Therefore, social media usage moderates the relationship between knowledge-sharing and

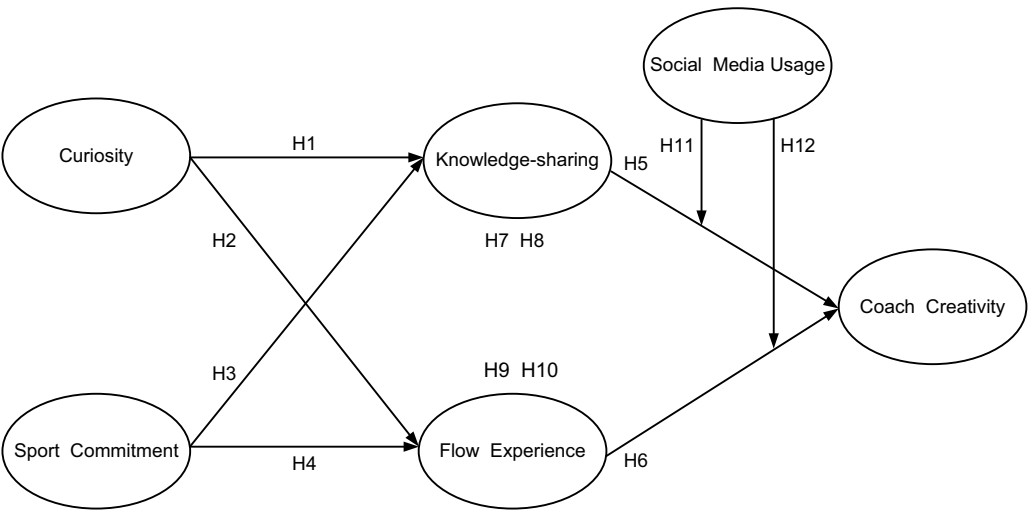

**Figure 1** Research model.

creativity. In the study, we hypothesized that greater use of social media by fitness coaches would have an impact on the relationship between knowledge-sharing and creativity. Hence, the hypothesis is as follows:

**H11:** The level of social media usage by fitness coaches will moderate the impact of knowledge-sharing on creativity.

Studies have shown that social media usage has an impact on the flow experience. *Leung (2020)* surveyed 653 mobile phone users and investigated how the flow experience and different mobile phone activities influence how bored people feel during their free time. Research has found that people are more likely to reach a state of flow when they use their smartphones to socialize. Additionally, *Zhao & Wagner (2022)* explore the main mechanism of short video platforms for user flow experience. Research shows that three types of technical support help TikTok users get optimal flow experience.

In summary, social media usage has an impact on the flow experience and creativity. Therefore, social media usage has a moderating effect on the link between flow experience and creativity. In this study, it is hypothesized as follows:

**H12:** The level of social media usage by fitness coaches will moderate the impact of flow experience on creativity.

In short, we propose a research model as Fig. 1 shown.

# MATERIALS & METHODS

## Participants and data collection

Before data collection, the researchers obtained ethical approval from Zhangzhou Hospital Affiliated with Fujian Medical University (IRB No. 2024LWB 336). All participants provided informed consent online through Wenjuanxing (http://www.wjx.cn) before completing the questionnaire. The consent form clearly explained the study's purpose, procedures, confidentiality, anonymity, participants' rights, and their freedom to withdraw at any time

without consequences. The data collection process strictly adhered to ethical guidelines, and all collected data were used exclusively for academic research. This study examines the impact, mechanisms, and moderating effects of curiosity and sport commitment on fitness coaches' creativity across emerging venues such as gyms, fitness studios, online platforms, and personal training services. According to the 2022 "China Fitness Industry Data Report", China has 85,149 fitness facilities, serving 71.45 million people, with a fitness penetration rate of 5.06%. There are approximately 822,000 fitness coaches in China, 81.32% of whom are male (*CFIDR, 2022*).

An online survey was distributed through social media channels targeting Chinese fitness coaches. Wenjuanxing was used to host the questionnaire and collect responses between September 18 and October 1, 2024, yielding a total of 732 valid responses. All personal information was kept confidential, and no identifying data were disclosed in any reports. According to the sample size calculator from SurveySystem (https://www.surveysystem.com), the collected sample meets the statistical requirements at a 95% confidence level with a 5% margin of error, ensuring population-level representativeness. In addition, based on recommendations by *Hair et al. (2017)*, a sample size of 200 or more is generally considered sufficient for partial least squares structural equation modeling (PLS-SEM) analysis, depending on model complexity. Our sample of 732 respondents meets the recommended thresholds, providing sufficient support for the robustness of the structural model estimation.

## Description of research variables

This study includes the variables of curiosity, sport commitment, knowledge-sharing, flow experience, coach creativity, social media usage, and basic background information of fitness coaches. The basic background information of fitness coaches, including gender, age, seniority, and monthly income, was collected based on the 2022 "China Fitness Industry Data Report" (*CFIDR, 2022*). For the six research variables mentioned above, a Likert scale with seven levels was used for measurement, where "1" indicates "strongly disagree" and "7" indicates "strongly agree." Higher scores indicate a higher level of agreement with the research variables. In addition, experts and academics were consulted for their input on the design of the questionnaire items. The descriptions of each construct in the questionnaire are as follows:

### Curiosity

The curiosity constructs in this study are based on *Tsai & Zheng (2021)*, research investigating the effects of different curiosity traits on service creativity among frontline service workers. Given the relevance of their studies' context to the current research context, this study adopts the same four dimensions to measure the curiosity of fitness coaches. Firstly, five items are used to measure joyous exploration, reflecting the fitness coach's tendency to view challenges as opportunities for growth and learning, enjoyment in acquiring new information, exploring unfamiliar topics, and engaging in deep thinking about certain questions, for example, "I view challenging situations as opportunities for growth and learning." Secondly, five items are used to measure deprivation sensitivity,

reflecting the fitness coaches' willingness to invest time and effort to solve problems and bridge knowledge gaps when faced with challenges, for example, "If I cannot find the answer to a problem, I feel frustrated and therefore work harder to solve it." Additionally, five items are used to measure social curiosity, reflecting the fitness coaches' interest in understanding others' habits, behavior patterns, conversational content, listening to their conversations, and curiosity about the reasons behind conflicts, for example, "I enjoy learning about other people's habits." Finally, five items are used to measure thrill-seeking, reflecting the fitness coaches' excitement toward novel experiences, for example, "The freshness of trying new things makes me feel excited and energized."

## Sport commitment

The construct of sport commitment in this research is based on the study by *Scanlan et al. (2016)*, which examined the second version of the Sport Commitment Scale's psychometric properties among 1,735 adolescent athletes aged 13–19 years. The study showed that the measure had good validity and reliability. Importantly, the scale can be administered as a whole, or specific subscales can be selected as needed. Based on their research, the Sport Commitment Scale consists of 12 sub-dimensions. Given the applicability of this scale to the current research context, this research adopts the enthusiastic commitment, constrained commitment, and sport enjoyment dimensions as the basis for measuring sport commitment. Firstly, six items are used to measure enthusiastic commitment, which primarily reflects the positive or "want to" aspect of commitment, for example, "I am committed to continuing fitness activities." Secondly, five items are used to measure constrained commitment, which primarily explains the reasons why athletes persist in their sport, for example, "Staying committed to fitness feels more like a need than a desire." Finally, five items are used to measure sport enjoyment, indicating the participants' level of enjoyment and positive emotions associated with their involvement in the sport, for example, "Fitness activities are fun."

## Knowledge-sharing

Knowledge-sharing as a construct in this study is built on the research of *Kim & Lee (2013)*, examining the connection between the goal orientation of knowledge-sharing and the service innovation behavior of employees in 5-star hotels in South Korea. The scale developed in their study has been widely used to measure knowledge-sharing behavior among employees in various organizations. Firstly, knowledge-collecting consists of four items that primarily reflect employees' tendency to seek information or knowledge by asking colleagues and their appreciation for colleagues sharing what they know, for example, "When I need certain knowledge, I consult my colleagues." Secondly, knowledge-donating also consists of four items that primarily reflect employees' inclination to share or inform colleagues when they learn something new and to update colleagues about their activities, for example, "When I learn something new, I take the initiative to share it with my colleagues."

## Flow experience

The flow experience measurement in this research is built upon the research of *Jackson & Marsh (1996)*, who developed and validated a flow experience scale among 394 athletes. The results show that the scale has good reliability and validity. The scale consists of nine dimensions, with each dimension measured by four items, resulting in a total of 36 items. Given the relevance of the scale to the context of this study, five dimensions were selected: unambiguous feedback, clear goals, concentration on the task at hand, transformation of time, and autotelic experience. Firstly, unambiguous feedback comprises four items that primarily reflect the perception of improvement, performance, and level experienced by fitness coaches during their teaching sessions, for example, "Every time I overcome obstacles in my training sessions, I know I've improved." Secondly, clear goals consist of four items that primarily reflect the goals of the coaching sessions and areas of the body that require enhanced training, for example, "I have a clear understanding of why I teach." Thirdly, concentration on the task at hand comprises four items that primarily reflect the degree of complete focus and attention experienced by fitness coaches during their teaching sessions, for example, "I can frequently become fully focused during my classes." Fourthly, the transformation of time consists of four items that primarily reflect the perceived change in the speed of time experienced by fitness coaches during their teaching sessions, for example, "When I am teaching, the passage of time seems different from usual." Lastly, autotelic experience is measured by four items, reflecting the positive feelings brought about by the instructor during teaching, including increased self-confidence, a pleasant mood, a sense of satisfaction, and the ability to relax both physically and mentally, for example, "I really enjoy the positive feelings brought by teaching."

## Coach creativity

Measuring coach creativity in this research is built on research conducted by *Janssen & Van Yperen (2004)*, who studied 170 employees in Dutch companies to explore the relationship between goal orientation, leadership, quality of communication between members, job performance, and job satisfaction. In their study, they used a scale consisting of 10 items to measure job creativity. Considering the relevance of their research to the context of this study and excluding reverse-scored items and items unrelated to this study, we adopted nine items to measure coach creativity. These items primarily reflect the ability of coaches to generate innovative ideas when faced with problems and to gain recognition from their colleagues for their innovative ideas, for example, "I propose new ideas to improve the situation."

## Social media usage

Measuring social media usage is based on research conducted by *Ali-Hassan, Nevo & Wade (2015)*, who conducted a study of employees in information technology companies, looking at the impact of social media usage on job performance. Considering the relevance of their research to the context of this study, we adopted five items from their study to measure social media usage. These items primarily reflect the level to which coaches utilize social media to make new social connections, maintain close social relationships, find

 

like-minded individuals, and create and disseminate work-related content, for example, "I use social media platforms like Douyin, WeChat, QQ, and Weibo to build new professional relationships at work."

## Data analysis

In this research, descriptive statistics were first conducted on the sample data. Subsequently, PLS-SEM was then applied, following the two-stage analysis approach proposed by *Anderson & Gerbing (1988)*. The first stage involved assessing the measurement model (out model), which included testing for convergent validity and discriminant validity. The second stage focused on evaluating the structural model (inner model), which involved examining the model fit and testing the proposed hypotheses, including direct effects, mediating effects, and moderating effects. Descriptive statistics were performed using IBM SPSS Statistics (version 26.0), while data analysis for the measurement model and structural model was conducted using Smart PLS 4.0 software.

# RESULTS

## Pilot test

Prior to the formal survey, a small-scale pilot test was conducted with 50 fitness instructors in China to assess the clarity, contextual appropriateness, and psychometric performance of adapted items. The pilot test included item-total correlation, item discrimination ($t$-tests), and internal consistency reliability (Cronbach's $\alpha$), serving to preliminarily evaluate the reliability and validity of the scales within this cultural and professional setting. All analyses were conducted using IBM SPSS Statistics (version 26.0).

As shown in Table 1, the item-total correlation coefficients ranged from 0.530 to 0.930, with most values exceeding 0.60, indicating that the items effectively measured their respective constructs. According to *DeVellis & Thorpe (2021)*, a coefficient of at least 0.30 is recommended; all items in this study substantially exceeded this threshold. In addition, the item discrimination analysis showed that $t$-values ranged from 3.139 to 7.779, demonstrating that the items had good discriminative power in distinguishing participants at different levels of the measured traits. According to *Field (2024)*, a $t$-value greater than 3.0 is generally considered indicative of strong item discrimination, and all items in this study met or exceeded this criterion. Finally, the Cronbach's $\alpha$ coefficients for all constructs ranged from 0.833 to 0.950, indicating excellent internal consistency. According to *Nunnally & Bernstein (1994)*, an $\alpha$ coefficient of at least 0.70 is considered acceptable for psychological measurements; all constructs in this study surpassed this benchmark.

## Descriptive statistics

The study collected data from fitness coaches employed by Chinese fitness companies. Based on Table 2, the study collected data from fitness coaches employed by Chinese fitness companies. The results indicate a total of 732 participants in the survey. In terms of gender, 70.6% were male ($n = 517$), while the remaining 29.4% were female ($n = 215$). Regarding age, 91% of coaches were 21–35 years old, indicating a relatively young population of

**Table 1  Results of the pilot test analysis.**

| Construct | Item-total correlation | Item discrimination | Cronbach's α |
|---|---|---|---|
| Joyous exploration | 0.811–0.875 | 5.497–7.779 | 0.941 |
| Deprivation sensitivity | 0.573–0.685 | 3.556–6.496 | 0.841 |
| Social curiosity | 0.530–0.820 | 3.547–7.609 | 0.850 |
| Thrill-seeking | 0.543–0.790 | 3.542–6.188 | 0.844 |
| Enthusiastic commitment | 0.724–0.891 | 3.540–5.667 | 0.943 |
| Constrained commitment | 0.657–0.794 | 3.417–6.733 | 0.878 |
| Sport enjoyment | 0.798–0.900 | 4.680–6.298 | 0.941 |
| Knowledge-collecting | 0.829–0.907 | 4.999–5.365 | 0.948 |
| Knowledge-donating | 0.625–0.731 | 3.139–6.080 | 0.833 |
| Unambiguous feedback | 0.780–0.841 | 5.037–6.375 | 0.919 |
| Clear goals | 0.695–0.803 | 5.555–6.686 | 0.896 |
| Concentration on task at hand | 0.764–0.864 | 4.907–5.978 | 0.924 |
| Transformation of time | 0.642–0.804 | 4.099–4.995 | 0.855 |
| Autotelic experience | 0.609–0.930 | 5.247–6.258 | 0.913 |
| Coach creativity | 0.667–0.862 | 4.275–5.592 | 0.950 |
| Social media usage | 0.657–0.826 | 4.379–5.342 | 0.898 |

coaches. In terms of tenure, 24.5% had 2–3 years of experience, 27.2% had 4–5 years, and 28.7% had 6–10 years. The distribution of tenure across different ranges is reasonable, especially with 13.3% having less than 1 year of experience, suggesting a competitive environment and reasonable mobility among coaches. Concerning monthly income (in RMB), 25% earned between 5,001–10,000, 25.3% earned between 10,001–15,000, and 29.2% earned between 15,001–30,000.

In addition to demographic characteristics, descriptive statistics for all measurement items were calculated. The mean scores ranged from 3.760 to 6.250, and standard deviations ranged from 0.985 to 2.174. Table 3 presents the mean and standard deviation of each item, including curiosity, sport commitment, knowledge-sharing, flow experience, coach creativity, and social media usage.

## Common method bias

This study employed *Harman*'s (*1976*) single-factor test to examine common method bias (CMB), and six factors were extracted. If the percentage of variance explained by the squared loading of the first factor's total variance is less than 50%, it indicates the absence of CMB. The results indicated that the largest variable variance explained percentage in this study was 39.16%, which met the criteria. Therefore, CMB is not a significant concern in this study.

## Measurement model results

The outer model of this study is a reflective measurement model, which assesses the reliability and validity of the measurement model for convergent validity. This includes item reliability (standardized factor loadings), construct reliability (Cronbach's α and

**Table 2  Sample demographics.**

| Demographics | Category | Frequency ($n = 732$) | Percent (%) |
|---|---|---|---|
| Gender | Male | 517 | 70.6 |
| | Female | 215 | 29.4 |
| Age (years) | ≤20 | 9 | 1.2 |
| | 21–25 | 212 | 29 |
| | 26–30 | 273 | 37.3 |
| | 31–35 | 181 | 24.7 |
| | 36–40 | 41 | 5.6 |
| | >40 | 16 | 2.2 |
| Seniority (years) | ≤1 | 97 | 13.3 |
| | 2–3 | 179 | 24.5 |
| | 4–5 | 199 | 27.2 |
| | 6–10 | 210 | 28.7 |
| | >10 | 47 | 6.4 |
| Monthly Income (RMB) | ≤5,000 | 75 | 10.2 |
| | 5,001–10,000 | 183 | 25 |
| | 10,001–15,000 | 185 | 25.3 |
| | 15,001–30,000 | 214 | 29.2 |
| | >30,000 | 75 | 10.2 |

composite reliability), as well as average variance extracted (AVE). To evaluate discriminant validity between constructs, this study applied two complementary approaches. Firstly, following the Fornell-Larcker criterion, the discriminative validity between constructs is evaluated using the AVE method. Secondly, in line with the recommendations of *Hair et al. (2019)*, the Heterotrait-Monotrait (HTMT) ratio was applied, which provides a more rigorous evaluation of discriminant validity.

## Convergent validity

In this study, standardized factor loadings are used to estimate the item reliability of the measurement model. *Fornell & Larcker (1981)* and *Nunnally (1978)* suggested that factor loadings higher than 0.7 indicate high item reliability. In Table 4, the factor loadings of each item in this study range from 0.718 to 0.946, all exceeding 0.7. Therefore, this shows that all measurement items have high item reliability.

In this study, Cronbach's $\alpha$ and composite reliability (CR) are employed to assess the internal consistency of constructs. *Nunnally (1978)* recommended that both Cronbach's $\alpha$ and the CR value should be above 0.70. In this study, as shown in Table 4, the Cronbach's $\alpha$ values for all constructs range from 0.745 to 0.960, and the CR values range from 0.857 to 0.969, all surpassing 0.7. Therefore, it can be concluded that all constructs demonstrate high levels of internal consistency reliability.

This study employed average variance extracted (AVE) to assess the convergent validity of the measurement model. *Fornell & Larcker (1981)* suggested that AVE should be greater than 0.5. In Table 4, AVE values for this study range from 0.581 to 0.862, indicating that the

**Table 3  Descriptive statistics of measurement items.**

| Item | Mean | Standard deviation | Item | Mean | Standard deviation |
|---|---|---|---|---|---|
| JE1 | 6.160 | 1.116 | COC4 | 5.520 | 1.263 |
| JE2 | 6.050 | 1.160 | COC5 | 5.800 | 1.160 |
| JE3 | 5.980 | 1.091 | COC6 | 5.680 | 1.197 |
| JE4 | 6.200 | 1.054 | COC7 | 5.570 | 1.265 |
| JE5 | 6.060 | 1.068 | COC8 | 5.640 | 1.248 |
| DS1 | 5.390 | 1.431 | COC9 | 5.490 | 1.319 |
| DS2 | 5.590 | 1.351 | KC1 | 5.970 | 1.154 |
| DS3 | 5.440 | 1.346 | KC2 | 6.030 | 1.108 |
| DS4 | 6.070 | 1.111 | KC3 | 6.100 | 1.084 |
| DS5 | 6.110 | 0.985 | KC4 | 6.040 | 1.116 |
| SC1 | 5.330 | 1.491 | KD1 | 5.700 | 1.254 |
| SC2 | 5.200 | 1.592 | KD2 | 5.720 | 1.307 |
| SC3 | 4.730 | 1.771 | KD3 | 5.120 | 1.573 |
| SC4 | 5.360 | 1.552 | KD4 | 4.780 | 1.688 |
| SC5 | 4.950 | 1.562 | UF1 | 5.910 | 1.163 |
| TS1 | 5.710 | 1.286 | UF2 | 5.980 | 1.110 |
| TS2 | 4.810 | 1.671 | UF3 | 5.880 | 1.175 |
| TS3 | 3.760 | 1.988 | UF4 | 5.920 | 1.077 |
| TS4 | 4.260 | 1.969 | CG1 | 6.050 | 1.058 |
| TS5 | 5.110 | 1.542 | CG2 | 5.920 | 1.106 |
| EC1 | 6.130 | 1.143 | CG3 | 5.890 | 1.194 |
| EC2 | 6.010 | 1.114 | CG4 | 5.920 | 1.121 |
| EC3 | 6.080 | 1.121 | COTAH1 | 5.860 | 1.151 |
| EC4 | 6.130 | 1.123 | COTAH2 | 5.850 | 1.266 |
| EC5 | 6.250 | 1.009 | COTAH3 | 5.940 | 1.118 |
| EC6 | 5.560 | 1.416 | COTAH4 | 6.000 | 1.044 |
| CC1 | 6.020 | 1.111 | TOT1 | 5.950 | 1.101 |
| CC2 | 5.240 | 1.767 | TOT2 | 5.380 | 1.511 |
| CC3 | 4.640 | 2.174 | TOT3 | 5.860 | 1.157 |
| CC4 | 5.900 | 1.496 | TOT4 | 5.440 | 1.453 |
| CC5 | 5.490 | 1.758 | AE1 | 5.950 | 1.136 |
| SPE1 | 6.090 | 1.171 | AE2 | 6.080 | 1.069 |
| SPE2 | 6.140 | 1.072 | AE3 | 6.060 | 1.063 |
| SPE3 | 6.100 | 1.138 | AE4 | 5.850 | 1.210 |
| SPE4 | 6.180 | 1.060 | SMU1 | 5.670 | 1.318 |
| SPE5 | 6.180 | 1.097 | SMU2 | 5.660 | 1.308 |
| COC1 | 5.890 | 1.122 | SMU3 | 5.530 | 1.381 |
| COC2 | 5.470 | 1.365 | SMU4 | 5.590 | 1.369 |
| COC3 | 5.930 | 1.098 | SMU5 | 5.670 | 1.326 |

**Notes.**

JE, Joyous exploration; DS, Deprivation sensitivity; SC, Social curiosity; TS, Thrill-seeking; EC, Enthusiastic commitment; CC, Constrained commitment; SPE, Sport enjoyment; KC, Knowledge-collecting; KD, Knowledge-donating; UF, Unambiguous feedback; CG, Clear goals; COTAH, Concentration on task at hand; TOT, Transformation of time; AE, Autotelic experience; COC, Coach creativity; SMU, Social media usage.

**Table 4 Item reliability, construct reliability, and convergent validity.**

| Construct | Item reliability | Construct reliability | | Convergent validity |
|---|---|---|---|---|
| | Factor loading | Cronbach's $\alpha$ | CR | AVE |
| Joyous exploration | 0.784–0.856 | 0.875 | 0.909 | 0.667 |
| Deprivation sensitivity | 0.718–0.829 | 0.819 | 0.874 | 0.581 |
| Social curiosity | 0.731–0.792 | 0.833 | 0.882 | 0.598 |
| Thrill-seeking | 0.731–0.839 | 0.822 | 0.874 | 0.582 |
| Curiosity | 0.728–0.843 | 0.777 | 0.857 | 0.601 |
| Enthusiastic commitment | 0.753–0.917 | 0.934 | 0.949 | 0.755 |
| Constrained commitment | 0.759–0.873 | 0.858 | 0.903 | 0.700 |
| Sport enjoyment | 0.900–0.946 | 0.960 | 0.969 | 0.862 |
| Sport commitment | 0.724–0.926 | 0.786 | 0.876 | 0.704 |
| Knowledge-collecting | 0.860–0.918 | 0.920 | 0.944 | 0.807 |
| Knowledge-donating | 0.806–0.872 | 0.865 | 0.908 | 0.712 |
| Knowledge-sharing | 0.899–0.901 | 0.745 | 0.887 | 0.796 |
| Unambiguous feedback | 0.878–0.905 | 0.914 | 0.940 | 0.796 |
| Clear goals | 0.867–0.900 | 0.904 | 0.933 | 0.777 |
| Concentration on task at hand | 0.872–0.917 | 0.919 | 0.943 | 0.805 |
| Transformation of time | 0.748–0.877 | 0.832 | 0.888 | 0.665 |
| Autotelic experience | 0.854–0.932 | 0.926 | 0.948 | 0.820 |
| Flow experience | 0.859–0.914 | 0.934 | 0.950 | 0.793 |
| Coach creativity | 0.796–0.882 | 0.951 | 0.958 | 0.720 |
| Social media usage | 0.921–0.923 | 0.824 | 0.919 | 0.850 |

**Notes.**
CR, Composite reliability; AVE, Average variances extracted.

AVE values for all constructs are greater than 0.5. Thus, all constructs exhibit convergent validity.

## Discriminant validity

This study employed the Fornell-Larcker criterion as an indicator to assess the discriminant validity of the measurement model, which suggests that the square root of AVE for each construct should be greater than the correlation coefficients between that construct and other constructs in the model (*Fornell & Larcker, 1981*). This demonstrates that each construct exhibits discriminant validity. In Table 5, the square roots of AVE for each construct in this study are all greater than their correlation coefficients with other constructs, thus confirming the discriminant validity of the measurement model in this study.

Furthermore, to strengthen the assessment of discriminant validity, this study also employed the HTMT ratio, a more rigorous criterion recommended by *Hair et al. (2019)*. The HTMT threshold is 0.90, indicating that values below this limit confirm adequate discriminant validity. As shown in Table 6, all HTMT values in this study fall below the recommended threshold, providing additional evidence that the constructs exhibit sufficient discriminant validity.

**Table 5 Fornell-Larcker criterion for discriminant validity.**

| Construct | Coach creativity | Social media usage | Curiosity | Flow experience | Knowledge-sharing | Sport commitment |
|---|---|---|---|---|---|---|
| Coach creativity | **0.848** | | | | | |
| Social media usage | 0.619 | **0.922** | | | | |
| Curiosity | 0.573 | 0.471 | **0.775** | | | |
| Flow experience | 0.790 | 0.616 | 0.609 | **0.891** | | |
| Knowledge-sharing | 0.699 | 0.582 | 0.613 | 0.760 | **0.892** | |
| Sport commitment | 0.566 | 0.480 | 0.583 | 0.689 | 0.605 | **0.839** |

**Notes.**
The bold values show the square root of the AVE, while the other matrix entries show the correlations.

**Table 6 HTMT ratio for discriminant validity.**

| Construct | Coach creativity | Curiosity | Flow experience | Knowledge-sharing |
|---|---|---|---|---|
| Coach creativity | | | | |
| Curiosity | 0.613 | | | |
| Flow experience | 0.820 | 0.650 | | |
| Knowledge-sharing | 0.752 | 0.680 | 0.812 | |
| Sport commitment | 0.597 | 0.632 | 0.725 | 0.662 |

## Construct model results
### Goodness of fit

To assess the quality of the proposed model, this study employed goodness of fit (GOF) as an assessment criterion. According to *Vinzi, Trinchera & Amato (2010)*, GOF is a value ranging from 0 to 1, with three levels of fit: weak (0.1), moderate (0.25), and strong (0.36) (*Wetzels, Odekerken & Van, 2009*). The calculation of GOF, as proposed by *Tenenhaus et al. (2005)*, is as follows:

$$\text{GOF} = \sqrt{\overline{\text{AVE}} \times \overline{R^2}} = \sqrt{0.744 \times 0.560} = 0.645.$$

In this study, the AVE (Table 4) and $R^2$ (Fig. 2) were included in this equation. The results indicated that the model's GOF was 0.645, which is higher than 0.36. This suggests that the model has a strong fit and is considered acceptable.

## Direct effects

In this research, the PLS-SEM analysis technique was used to evaluate the inner model. The main indicators include the significance testing of path coefficients and the $R^2$ values (assessing the magnitude of $R^2$). Path coefficients refer to the strength and direction of relationships between variables, indicating the influence of predictors on the dependent variable. More specifically, they represent the research hypotheses related to the direct effects of this study. Additionally, $R$-squares represent the percentage of variance explained by the dependent variable, indicating the predictive capability of the model. Finally, the

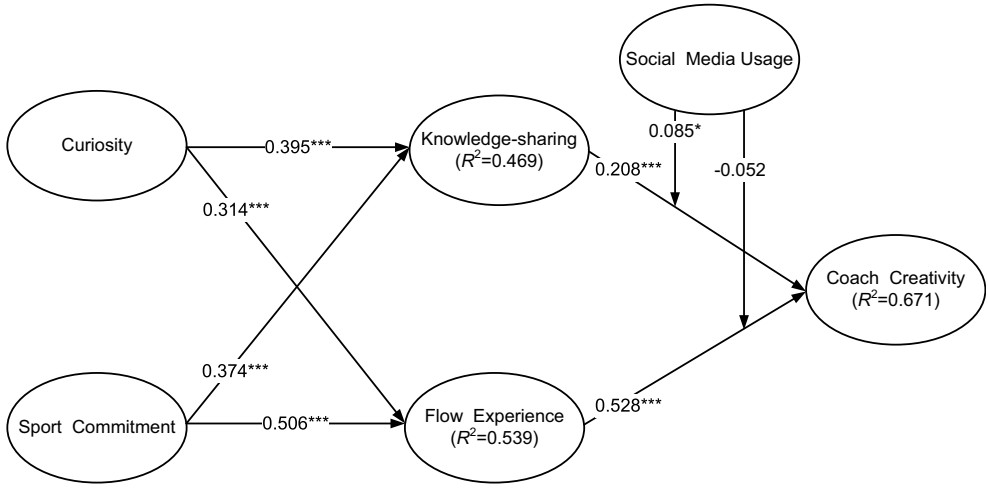

*p<0.05, *** p<0.001

**Figure 2  Test result of research hypothesis.**

**Table 7  Results for hypothesis testing (H1–H6).**

| Hypothesis | Relationship | Original sample | Standard deviation | T statistics | P values | $R^2$ |
|---|---|---|---|---|---|---|
| H1 | CUR → KS | 0.395 | 0.038 | 10.490 | 0.000 | 0.469 |
| H3 | SC→ KS | 0.374 | 0.037 | 10.124 | 0.000 | |
| H2 | CUR → FE | 0.314 | 0.041 | 7.696 | 0.000 | 0.539 |
| H4 | SC→ FE | 0.506 | 0.036 | 14.182 | 0.000 | |
| H5 | KS→COC | 0.208 | 0.041 | 5.116 | 0.000 | 0.671 |
| H6 | FE→ COC | 0.528 | 0.044 | 11.877 | 0.000 | |

**Notes.**
CUR, Curiosity; KS, Knowledge-sharing; FE, Flow experience; SC, Sport commitment; COC, Coach creativity.

Smart PLS 4.0 software was used to analyze the model. Through path coefficients, we obtained standard beta ($\beta$), $t$-value, $p$-value, and the coefficient of determination ($R^2$).

In Table 7, the findings show that: firstly, curiosity and sport commitment have a significant positive impact on knowledge-sharing, supporting H1 (CUR→KS: $\beta = 0.395$, $t = 10.49$, $p < 0.001$) and H3 (SC→KS: $\beta = 0.374$, $t = 10.124$, $p < 0.001$). Secondly, curiosity and sport commitment have a significant positive impact on flow experience, supporting H2 (CUR→FE: $\beta = 0.314$, $t = 7.696$, $p < 0.001$) and H4 (SC→FE: $\beta = 0.506$, $t = 14.182$, $p < 0.001$). Finally, knowledge-sharing and flow experience have a significant positive impact on coach creativity, supporting H5 (KS→COC: $\beta = 0.208$, $t = 5.116$, $p < 0.001$) and H6 (FE→COC: $\beta = 0.528$, $t = 11.877$, $p < 0.001$).

An $R^2$ value greater than 0.67 indicates high explanatory power, around 0.33 indicates moderate explanatory power and around 0.19 indicates low explanatory power (*Chin, 1998*). In Table 7, the results reveal that the $R^2$ value for knowledge-sharing is 0.469, for flow experience is 0.539, and for coach creativity is 0.671. These results indicate

**Table 8  Results for mediating effects (H7–H10).**

| Hypothesis | Relationship | Bootstrapping ($n = 1,000$) | | | | 95% CI | | Result |
|---|---|---|---|---|---|---|---|---|
| | | Original sample | Standard deviation | T statistics | P values | 2.50% (Lower) | 97.50% (Upper) | |
| H7 | CUR→KS→COC | 0.082 | 0.018 | 4.486 | 0.000 | 0.049 | 0.121 | Supported |
| H8 | SC→KS→COC | 0.078 | 0.017 | 4.703 | 0.000 | 0.048 | 0.111 | Supported |
| H9 | CUR→FE→COC | 0.166 | 0.025 | 6.651 | 0.000 | 0.117 | 0.215 | Supported |
| H10 | SC→FE→COC | 0.267 | 0.030 | 8.844 | 0.000 | 0.211 | 0.324 | Supported |

Notes.
CUR, Curiosity; KS, Knowledge-sharing; FE, Flow experience; SC, Sport commitment; COC, Coach creativity; CI, Confidence interval.

that 46.9% of the variance in knowledge-sharing is attributed to curiosity and sport commitment. Furthermore, 53.9% of the variance in flow experience is caused by curiosity and sport commitment. Additionally, 67.1% of the variance in coach creativity is driven by knowledge-sharing and flow experience.

## Mediating effects

To assess the statistical significance of the proposed mediation models, this study conducted data analysis using Smart PLS 4.0 software. Specifically, bootstrapping with 1,000 resamples was employed to estimate the significance of each mediating effect. Bootstrapping is a robust method that involves repeatedly sampling from the dataset to obtain more accurate statistical results (*Purvis, Sambamurthy & Zmud, 2001*). In this analysis, mediating effects were considered significant if the confidence interval did not include zero. The results of the bootstrapping analysis, as presented in Table 8, support the presence of four mediating effects.

Firstly, curiosity exerts a positive mediating effect on coach creativity through knowledge-sharing (CUR→KS→COC: CI [0.049–0.121], $p < 0.001$), with a confidence interval excluding zero, supporting H7. Secondly, sport commitment exerts a positive mediating effect on coach creativity through knowledge-sharing (SC→KS→COC: CI [0.048–0.111], $p < 0.001$), with a confidence interval excluding zero, supporting H8. Thirdly, curiosity exerts a positive mediating effect on coach creativity through flow experience (CUR→FE→COC: CI [0.117–0.215], $p < 0.001$), with a confidence interval excluding zero, supporting H9. Lastly, sport commitment exerts a positive mediating effect on coach creativity through flow experience (SC→FE→COC: CI [0.211–0.324], $p < 0.001$), with a confidence interval excluding zero, supporting H10. In summary, the results indicate that both knowledge-sharing and flow experience play significant mediating roles in the relationship between the predictors (curiosity and sport commitment) and the outcome variable (coach creativity), thus supporting H7, H8, H9, and H10.

## Moderating effects

To evaluate the statistical significance of the proposed moderation effects, this study followed the approach suggested by *Aiken, West & Reno (1991)*. Smart PLS 4.0 software was utilized for data analysis, and the significance of the moderating effects was determined

**Table 9   Results for moderating effects (H11–H12).**

| Hypothesis | Relationship | Original sample | Standard deviation | T statistics | P values | Result |
|---|---|---|---|---|---|---|
| H11 | SMU*KS→ COC | 0.085 | 0.042 | 2.050 | 0.041 | Supported |
| H12 | SMU*FE→ COC | −0.052 | 0.038 | 1.372 | 0.170 | Not Supported |

**Notes.**

SMU, Social media usage; KS, Knowledge-sharing; FE, Flow experience; COC, Coach creativity.

through bootstrapping. If the *p*-value is less than 0.05, the moderating effect is considered significant.

In Table 9, social media usage positively moderates the impact of knowledge-sharing on coach creativity, thus supporting H11 (SMU*KS→COC: $t = 2.050 > 1.96$, $p < 0.05$). However, the moderating effect of social media usage on the relationship between flow experience and coach creativity is not significant, therefore not supporting H12 (SMU*FE→COC: $t = 1.372 < 1.96$, $p > 0.05$).

## DISCUSSION

### Key findings

This study aims to understand the formation mechanism of fitness coach creativity. Our research results indicate that knowledge-sharing and flow experience mediate the relationship between curiosity and sport commitment to creativity. Furthermore, social media usage positively moderates the relationship between knowledge-sharing and creativity. Some key findings are summarized as follows:

This study affirms the substantial influence of knowledge-sharing and flow experience on creativity. Initially, our findings indicate a positive and significant effect of knowledge-sharing on the creativity of fitness coaches. This finding resonates with prior research (*Tsai & Zheng, 2021*), underscoring the pivotal role of knowledge-sharing in nurturing creativity among professionals. Specifically, fitness coaches who willingly share their knowledge are more inclined to engage in discussions on various aspects of fitness programs with peers, within fitness communities, or on online platforms. Through knowledge-sharing, individuals can glean insights from diverse backgrounds and fields, fostering novel perspectives and creative ideation. Additionally, our research reveals that the flow experience significantly contributes to the creativity of fitness coaches. This aligns with *Schutte & Malouff (2020)* perspective on the correlation between flow experience and creativity in college students. It suggests that being in a state of flow, characterized by high engagement and concentration, is conducive to enhancing the creative abilities of fitness coaches.

Curiosity and sport commitment influence knowledge-sharing. Firstly, curiosity significantly impacts knowledge-sharing, consistent with the findings of *Tsai & Zheng (2021)*. In other words, the work of a fitness coach relies more on knowledge and skills. Curiosity prompts fitness coaches to actively share knowledge with others when faced with a specific task, allowing them to delve into a more comprehensive understanding of the task's content and requirements. This finding aligns with *Kashdan et al. (2020)* perspective,

which suggests that curiosity makes them more aware of knowledge gaps, driving them to actively share knowledge to maintain a competitive edge. Secondly, sport commitment has a significant impact on knowledge-sharing, in line with previous research examining the influence of different commitment types on knowledge-sharing (*Luo et al., 2021*; *Ouakouak & Ouedraogo, 2019*). This implies that when fitness coaches have higher levels of sport commitment, they are more willing to engage in knowledge-sharing in order to help clients achieve their fitness goals. This finding is consistent with the Sport Commitment Model's viewpoint, which states that sport commitment represents a professional trait of coaches (*Scanlan et al., 1993a*) and can influence their behavior (*Lukwu & Guzmán, 2011*).

Curiosity and sport commitment influence the flow experience. Firstly, curiosity significantly affects the flow experience, aligning with the findings of *Schutte & Malouff (2020)*. Our research indicates that when fitness coaches become curious about a particular task, they invest more energy and attention, making it easier for them to enter the flow state due to this heightened involvement. Secondly, sport commitment significantly impacts the flow experience, consistent with the previous studies on the influence of different types of commitment (*Cheng, Hung & Chen, 2016*). This finding is in line with *Scanlan et al. (1993a)* perspective that sport commitment signifies an individual's high involvement and sense of responsibility in a specific domain. This finding suggests that fitness coaches who demonstrate a greater level of commitment to their sport experience increased engagement and concentration, particularly when working towards fulfilling their clients' fitness objectives.

This study confirms the mediating effects of knowledge-sharing and flow experience. In other words, knowledge-sharing and flow experience serve as key mediators linking curiosity and sport commitment to creativity in the fitness coaching context. These traits promote active knowledge exchange and immersive engagement, which in turn enhance cognitive input and social validation-important resources that support the generation of novel and useful ideas. In line with prior research, curiosity influences creativity through knowledge-sharing (*Tsai & Zheng, 2021*) and flow experience (*Schutte & Malouff, 2020*). Furthermore, this study extends prior empirical findings by demonstrating that curiosity influences creativity *via* knowledge-sharing and flow experience in the fitness coaching context (*Celik et al., 2016*). This finding underscores the critical role of knowledge-sharing and flow experience in shaping creativity. Additionally, existing research has indicated the significant impact of commitment on creativity (*Jeon & Choi, 2020*). However, scholars have not explored knowledge-sharing and flow experience as mediating mechanisms in the relationship between sport commitment and creativity. This study confirmed the mediating roles of knowledge-sharing and flow experience. This novel insight emphasizes the importance of understanding the relationship between sport commitment and creativity. Notably, knowledge-sharing functions as a cognitive-social mechanism that facilitates the exchange of insights and best practices (*Kim & Lee, 2013*), while flow experience serves as a motivational mechanism that promotes deep focus and intrinsic enjoyment (*Csikszentmihalyi, 1990*). Together, these processes explain how internal traits are transformed into creative performance.

Furthermore, we found that sport commitment has the most significant mediating effect on creativity through flow experience. Firstly, concerning knowledge-sharing, the influence of curiosity is more important than sport commitment. It can be observed that when there is a demand for knowledge, curiosity can drive the inquiry, collection, and donation of knowledge. Similarly, in terms of flow experience, the influence of sport commitment is more significant than curiosity. This could be related to specific tasks such as fitness guidance, training programs, and other task-oriented contexts. In these task-oriented situations, individuals prioritize task completion and meeting customer needs (*Sánchez-Miguel et al., 2019*) rather than merely satisfying their curiosity. Therefore, fitness coaches with high sport commitment are better equipped to drive individuals to engage in flow experience. Secondly, when it comes to creativity, flow experience is more important than knowledge-sharing. According to the social information processing theory, external pressures can influence an individual's knowledge-sharing behavior (*Chen & Hung, 2010*). This implies that, in the face of competition, fitness coaches may be less inclined to share knowledge (*Kim & Lee, 2013*). However, flow experience is an intrinsic motivation that is less susceptible to external interference (*Csikszentmihalyi, 1990*). In a state of flow experience, fitness coaches are more likely to unleash their creativity. In summary, the sport commitment of fitness coaches significantly influences creativity through the flow experience.

Existing literature suggests that the use of social media is a significant factor influencing knowledge-sharing and creativity (*Zhang et al., 2021*). Building on this, our research findings indicate that social media usage positively moderates the relationship between knowledge-sharing and creativity. This novel insight suggests that through the use of social media, fitness coaches can access knowledge from different domains, thereby contributing to enhanced creative thinking. However, this study found that social media usage did not significantly moderate the relationship between flow experience and creativity. This result suggests that the impact of social media on creativity may depend on the nature of the creative process. Flow experience is an intrinsic motivational state, which may not be significantly influenced by external factors such as social media usage (*Amabile, 1983*; *Csikszentmihalyi, 1990*). Furthermore, excessive social media use may lead to information overload, potentially disrupting the deep engagement required for flow. Additionally, fitness coaches might primarily use social media for marketing or client engagement rather than as a source of creative inspiration. Future research should explore different types of social media usage and examine whether these have differential effects on creativity.

## Theoretical contributions

This study aims to increase the body of knowledge in this area due to the limited literature on fitness coaches. The research makes theoretical contributions by validating, extending, and expanding the knowledge in the following aspects.

Firstly, research in the field of fitness is predominantly conducted in the United States (*Kim & Byon, 2021*) and Spain (*García-Fernández et al., 2018*). In the context of China, there is a relative lack of research in the fitness industry. Furthermore, currently, scholars in the field of sports creativity have explored various aspects such as motor creativity (*Richard*

*et al., 2018*), and tactical creativity (*Furley & Memmert, 2018*). Moreover, the majority of fitness-related research focused on the consumer level (*Fernández-Martínez et al., 2020*) or managerial perspectives (*León-Quismondo, García-Unanue & Burillo, 2020*). There is a lack of empirical research from the perspective of fitness coaches. In particular, existing research has not focused on the factors influencing the creativity of fitness coaches and its significance. As frontline professionals, fitness coaches encounter various on-site or professional challenges. Given the high competition in the fitness industry in recent years and the adoption of innovative strategies, there is an increased reliance on the creativity of coaches, especially by those pursuing a differentiation competitive strategy.

Secondly, through empirical research, we explore the formation process of fitness coaches' creativity: curiosity and sport commitment enhance creativity by influencing knowledge-sharing and the flow experience, with the moderating effect of social media usage. Moreover, this study highlights the distinctive context of fitness coaching, characterized by continuous client interaction, real-time responsiveness, and personalized instruction, which uniquely shapes how knowledge-sharing and flow experience operate as mediating mechanisms in the creativity formation process. By validating these mediating effects specifically within the service-intensive fitness coaching context, the findings contribute context-specific insights to existing creativity research (*Luo et al., 2021*; *Schutte & Malouff, 2020*; *Tsai & Zheng, 2021*).

Notably, we found that the influence of sport commitment on creativity through flow experience was significantly stronger than the path through knowledge-sharing, indicating that flow plays a more critical mediating role in the creativity formation process within the service-intensive fitness coaching context. This finding aligns with the principles of social information processing theory (*Nonaka & Takeuchi, 1996*) and Flow Theory (*Csikszentmihalyi, 1990*), both of which emphasize that external interactions and intrinsic motivation are central mechanisms driving creativity. Our study further extends this theoretical perspective by highlighting that, in the service-intensive context of fitness coaching, intrinsic motivational pathways such as flow may exert a relatively stronger influence.

This study also aligns with *Nonaka (2009)* knowledge-creation model, which posits that professional knowledge-sharing can be transformed into innovative outcomes. By emphasizing how professional knowledge-sharing in a service-oriented sports context drives creativity, this study expands the applicability of the knowledge-creation model to fitness coaches, enriching the understanding of creativity in this unique context. Furthermore, it validates and extends flow theory by demonstrating that individuals in a flow state exhibit significantly enhanced creativity, confirming its relevance in a service-oriented sports context and expanding its application in creativity research. Furthermore, this study emphasizes the boundary role of social media usage in the impact of fitness coaches' knowledge-sharing on creativity. It highlights the significance of social media usage in the era of the internet. The results confirm the excellent explanatory power of the proposed model, providing new and expansive evidence in the field of creativity. As fitness coaches' creativity becomes a competitive advantage for enterprises, research on this important topic holds significant empirical value.

Finally, by underscoring the pivotal role of sport commitment in predicting the emergence of creativity, this study not only contributes to the enriched understanding of creativity formation but also addresses a notable gap in existing research. While previous studies have predominantly delved into factors such as personality traits (*Amin et al., 2020*), goal orientation (*Zhang et al., 2020*), trust (*Chen et al., 2021*), the role of commitment has garnered limited attention. Therefore, this research serves to validate the significance of commitment in the genesis of creativity. Notably, the study extends the sport commitment model (*Scanlan et al., 1993a*) by revealing that sport commitment influences creativity indirectly through enhancing professional behaviors (*e.g.*, knowledge-sharing) and psychological states (*e.g.*, flow experience). At the same time, it highlights sport commitment as a stable professional trait that enables fitness coaches to remain engaged, focused, and innovation-driven in practice. This dual role strengthens the theoretical relevance of commitment and fills a gap in the creativity literature from the perspective of service professionals.

## Practical significance

This study provides compelling evidence that the creativity of fitness coaches should be given attention. The creativity of fitness coaches offers a new opportunity for fitness businesses to enhance their competitiveness. Therefore, it is crucial for managers and practitioners to better harness the creativity of fitness coaches.

Firstly, fitness companies can build upon this study to cultivate knowledge-sharing and flow experience among fitness coaches, which can contribute to enhancing creativity. To begin with, the research findings emphasize the critical importance of flow experience, and specific strategies should be implemented. Fitness companies should first establish clear goals and challenges for fitness coaches, facilitating focus and dedication to tasks. Moreover, fitness companies should ensure that the work environment and conditions allow fitness coaches to concentrate on their tasks and reduce external distractions. Additionally, fitness companies should provide adequate resources and support for fitness coaches, such as training programs and technical assistance, to ensure effective work engagement.

Secondly, fitness companies should also consider how to enhance knowledge-sharing among fitness coaches. In the face of intense competition in the fitness industry, knowledge-sharing can be influenced by external factors (*Chen & Hung, 2010*). The research findings underscore the importance of not only encouraging knowledge-sharing among fitness coaches but also giving due consideration to the use of social media. In practical terms, addressing this issue should involve several strategies. On one hand, fitness companies can establish a culture of proactive knowledge-sharing within their organization, create knowledge-sharing platforms, and improve reward and incentive mechanisms for knowledge-sharing. On the other hand, fitness companies can leverage social media as a tool for knowledge-sharing (*Chen, Kamalanon & Janupiboon, 2022b*), encouraging fitness coaches to create their own social media platforms such as Douyin, Kuaishou, Weibo, and others (*Kim & Jang, 2018*). Through social media usage, fitness coaches can share fitness techniques, workout plans, and more to expand their audience and increase opportunities for knowledge dissemination (*Shwartz-Asher et al., 2020*). This not only enhances customer

satisfaction but also boosts the overall creativity of fitness businesses (*Kim, Karatepe & Lee, 2019*; *Wibowo et al., 2020*). After achieving these goals, fitness companies can acquire new customers and revenue by recruiting members (*Kim & Byon, 2021*).

Furthermore, fitness companies should enhance knowledge-sharing and flow experience by recruiting fitness coaches who possess curiosity and sport commitment. If fitness coaches have low levels of sport commitment or curiosity, it will affect knowledge-sharing and flow experience. On one hand, fitness companies can employ high-commitment human resource management measures (*Mostafa et al., 2019*) to rigorously select fitness coaches with strong sport commitment. On the other hand, fitness companies can use personality trait assessments to prioritize the selection of fitness coaches with curiosity (*Kashdan et al., 2020*). This study proposes that curiosity and sport commitment are important considerations in recruiting fitness coaches, as both are inherent personal traits of coaches and should receive special attention in talent acquisition. Notably, given that sport commitment influences creativity most strongly through flow experience, fitness companies should also focus on designing work environments that facilitate flow; such as providing challenging tasks, clear goals, autonomy, and timely feedback to help coaches translate their commitment into creative performance.

Finally, as creativity plays a key role in driving innovation and competitiveness in the fitness industry, future efforts should also focus on the legal protection of creative outputs. Governments are encouraged to establish legal frameworks that ensure such outputs are safeguarded and applied within a compliant and regulated environment. Specifically, relevant intellectual property laws-such as patent law, copyright law, and trademark law-should be strengthened to ensure that creative outputs in the fitness industry receive legal recognition and exclusive protection. In addition, fitness enterprises should actively comply with legal regulations and adopt lawful means to protect their creative outputs, thereby preventing potential infringements. Governments are also advised to enhance international cooperation to ensure that creative outputs in the fitness industry are effectively protected and enforced on a global scale. This would not only help enterprises safeguard their unique creative outputs but also promote cross-border market expansion and international collaboration.

## Limitations and future research

Firstly, research on creativity remains a focal point of discussion among scholars (*Saleh & Brem, 2023*; *Scandle, Brown & Wagstaff, 2023*). The aim of this research is not to determine whether fitness coaches possess creativity, but rather to investigate how curiosity, sport commitment, and the mediating role of knowledge-sharing and flow experience affect the creativity of fitness coaches. Additionally, the moderating effect of social media usage is also discussed. Other factors that may influence creativity, such as self-efficacy and other personality traits, were not examined in this research. To explore this issue further, studies could look at some different influencing factors.

Secondly, this research used a cross-sectional design, which limits the ability to infer causal relationships between factors. Future research could utilize longitudinal analysis to examine relationships over different periods. Thirdly, this study focused on the mediating

roles of curiosity and sport commitment in enhancing creativity among Chinese fitness coaches. While the findings contribute to understanding psychological drivers in this specific cultural and occupational context, generalizing the model globally requires caution. Since cultural factors, coaching practices, and professional norms differ across regions, they may influence how curiosity and commitment function in various settings. Thus, future research should examine the model in diverse cultural and professional contexts to validate its broader applicability. Fourthly, although the model sheds light on individual mechanisms, it does not fully capture team-based influences. Further studies could incorporate team-level factors such as collective flow or team commitment.

Moreover, the study also did not differentiate types of social media usage, which may play varying roles in digital knowledge-sharing and creativity. Subgroup analysis by coaching type was omitted due to assumed psychological consistency across roles, but future studies may explore contextual differences. Lastly, validated scales were used and PLS-SEM, which does not require data normality (*Hair et al., 2019*), was adopted as the primary analytical method. Future research may further explore measurement invariance across groups using covariance-based structural equation modeling (CB-SEM).

## CONCLUSIONS

This study explores the impact of curiosity and sport commitment on the creativity of fitness coaches. The results show that knowledge-sharing and flow experience have a significant positive effect on creativity through these mediating variables. Social media usage positively moderates the relationship between knowledge-sharing and creativity, but has no significant effect on the relationship between flow experience and creativity. By revealing the mediating role of knowledge-sharing and flow experience, this study expands the understanding of the mechanisms influencing fitness coaches' creativity. However, the non-significant moderating effect of social media on flow experience suggests that its role in different contexts needs further exploration. Future research could focus on various social media usage patterns and their specific impacts on the development of creativity in fitness coaches. Moreover, based on the findings of this study, creativity in the fitness industry plays a critical role in enhancing coaches' professional competitiveness, improving client experiences, and driving the industry's digital transformation. However, the current legal and policy frameworks do not yet provide adequate protection for the creative outputs of fitness coaches. As a result, the industry faces challenges such as vulnerability to intellectual property infringement, overly restrictive competition, and underdeveloped mechanisms for protecting creative rights. These issues not only undermine coaches' motivation to innovate but may also hinder the industry's long-term sustainability. Therefore, greater attention should be given to the legal protection of creative outputs. It is recommended that governments strengthen relevant legal systems to ensure that creative contributions are appropriately utilized and effectively protected within a legal framework, thereby promoting continuous innovation and healthy development within the fitness sector.

### Funding

The authors received no funding for this work.

### Competing Interests

The authors declare there are no competing interests.

### Author Contributions

- Ying Chen conceived and designed the experiments, performed the experiments, analyzed the data, prepared figures and/or tables, authored or reviewed drafts of the article, and approved the final draft.
- Bin Chen conceived and designed the experiments, performed the experiments, analyzed the data, prepared figures and/or tables, authored or reviewed drafts of the article, and approved the final draft.

### Ethics

The following information was supplied relating to ethical approvals (i.e., approving body and any reference numbers):

This study was approved by Zhangzhou Hospital Affiliated to Fujian Medical University (2024LWB336).

### Data Availability

The raw data is available in the Supplemental Files and Zenodo: Chen, B. (2024). Exploring the Impact of Curiosity and Sport Commitment on Creativity among Fitness Coaches: The Mediating Role of Knowledge-Sharing and Flow Experience [Data set]. Zenodo. https://doi.org/10.5281/zenodo.13855427.

### Supplemental Information

Supplemental information for this article can be found online at http://dx.doi.org/10.7717/peerj.19684#supplemental-information.

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
