# Peer review of "Exploring the impact of curiosity and sport commitment on creativity among fitness coaches: the mediating role of knowledge-sharing and flow experience"

_PeerJ, doi:10.7717/peerj.19684_

## Round 0.1 · original submission · Major Revisions

Dear Authors

Two experts in the study field have reviewed the manuscript. Your study has been reviewed with comprehensive comments to improve the quality of the manuscript. I would encourage you to highlight the study's novelty as the point raised by the reviewers. We invite you to submit a revised version of the manuscript addressing the reviewers’ comments.

We look forward to receiving your revised manuscript.

Best regards

Yung-Sheng Chen, Ph.D.
Academic Editor

Reviewer 1 ·

Basic reporting

No comments

Experimental design

1.Sampling Method
a. Please justify why focusing solely on Chinese fitness coaches
b. You mentioned you use Sample size calculator from surveysystem. However, this is a study of the model. I suggest you cross-check with G*power or Kline's (2023) suggestion for sample size.

2.Instrument issue:
a. Sport commitment questionnaire-2: You mentioned that you adopt SCQ-2 enthusiastic commitment, constraint commitment and also sport enjoyment items in line 397-400. Based on my understanding, both enthusiastic commitment and constraint commitment belong to sport commitment dimensions, while sport enjoyment was category as FACTOR to sport commitment. There are other factors such as social support, desire to excel etc. Please justify why you group EC,CC and sport enjoyment as “sport commitment” in your study while not include other factors such as desire to exile or social support in the SCM model.
b. Justify why using 7 point likert scale while the original questionnaire (SCQ-2) uses 5 point likert scale.

Validity of the findings

Analysis:
1. Do you conduct a pilot test? As pilot test is an important procedure to check the appropriateness of instrument and process of data collection.
2. As I go through your manuscript and instrument, I found that you are adapting different instruments/questionnaires to conduct this study. Your questionnaire adopt a 7 likert scale. However some of the original questionnaire such as SCQ-2 adopt 5 point likert scale. Thus, I assume that you need to conduct a SEM study to confirm the psychometric properties of the instrument, follow by assessing direct effect before looking into the moderator effect.
3. Furthermore, please justify why you are not conducting normality test.
4. Discriminant Validity. You’ve used Fornell & Larcker 1981 for discriminant validity. I suggest you to use Heterotrait-Monotrait Ratio of Correlations (HTMT) for assessing discriminant validity as it is more sensitive in identifying problems with discriminant validity, especially in complex models like those analyzed with PLS-SEM (Henseler et al. 2015).

Findings
1. The manuscript finds no significant moderating effect of social media on the relationship between flow experience and creativity. This non-significant finding could be discussed more critically, offering possible explanations and implications.
2. The manuscript effectively reports the statistical results, but it would benefit from an explicit discussion of the best model. Which is the best model?
3. Identifying and discussing the strongest and their implications for theory and practice.
4. Expanding on how this model contributes to the theoretical understanding of creativity and sport commitment while addressing its limitations. Incorporating these elements would strengthen the discussion, providing a more integrated and impactful narrative around the study's findings.

Theoretical contributions
1. Line 681-692 “sound” problem statement. Please Integrate the findings with Existing Theories.
2. I suggest the author could include the theoretical significance in this section.
3. Since this study outline the importance of the mediating role of curiosity and sport commitment among Chinese fitness coaches, therefore, please critically analyze the influence / contribution of the model to Chinese coaches as well as globally.

Limitations
1. Did you perform pilot test and normality test? If not please justify here in the LIMITATIONS
2. Please justify why EFA and CFA is not included in this study. As concerned, this study adopt and adapt several items from different questionnaire in a new context (fitness coaches) and culture (China).

Additional comments

No comments.

Reviewer 2 ·

Basic reporting

The subject of the current manuscript, titled “Exploring the Impact of Curiosity and Sport Commitment on Creativity among Fitness Coaches: The Mediating Role of Knowledge-Sharing and Flow Experience,” is both intriguing and pertinent to the readership of this journal. This study tackles a significant and relevant issue concerning the interplay between curiosity, sport commitment, knowledge-sharing, flow experience and creativity. The appropriate methodology employed further enhances the study's contribution by offering a comprehensive perspective on the mechanisms that underlie creativity. The structure of present manuscript is well-organzied, research result is self-contained with hypothesis. Nevertheless, several questions regarding research question, methodolgy, and discussion remain that warrant further clarification.

Experimental design

The introduction offers a comprehensive background on the current topic, linking fitness coaching with creativity; however, the research gap remains unclear. The examination of mediation elucidates the reasons behind the influence of the independent variable (IV) on the dependent variable (DV), addressing the question about “why”. The relationship between the IV and DV should be confirmed prior to conducting the mediation analysis. Furthermore, when the author addresses the mediating roles of knowledge sharing and flow experience in the relationship among sports commitment, curiosity, and creativity, it is crucial to underscore the specific context of fitness coaching. Otherwise, existing studies conducted in non-fitness coaching contexts, such as those by Tsai & Zheng (2021), Luo et al. (2021), Schutte & Malouff (2020), and Cheng et al. (2016), have already explored the mediating roles of knowledge sharing and flow experience. This raises the question of why the author should re-examine these variables within the context of fitness coaching. Consequently, the arguments regarding “curiosity and sports commitment affect creativity in a sports setting” and “flow experience as well as knowledge-sharing serve as mediators in the relationship between curiosity, sports commitment, and creativity” need clarification.

On a different note, moderation analysis investigates the conditions under which the IV enhances or diminishes the DV, addressing the question about “when”. In other word, the inconsistent relationship (e.g., direction or intensity) between IV and DV should be revealed before moderation test. The introduction lacks a clear rationale for selecting social media as a moderating variable. Although its influence on knowledge sharing, flow experience and creativity is acknowledged, but the effect of moderator is on the relation between IV and DV, not on IV or DV itself. Therefore, a more robust theoretical justification regarding “mediator explain why IV affect DV” and “moderator identify when IV affect DV” would strengthen this aspect of the study.

Regarding the methodology, several questions remain:
1. Are there different segments of fitness coaches (e.g., personal trainers, gym-based coaches, online fitness instructors)?
2. Did author conduct data screen (Tabachnick & Fidell, 2019)?
3. The mean and standard deviation of each variable should be presented.
4. The significance of the correlation coefficient requires clarification.
5. What is the sample item for each measurement?
6. Several measurements author applied are English version, statistic evidence supports the generalizability across culture is warranted.
7. What SEM did author conduct? Covariance-based or variance-based? And why?

Validity of the findings

In terms of discussion, the author's discourse on the mediating relationship is generally appropriate. However, the content largely presents interpretations of direct effects rather than mediating effects. The latter should emphasize the logic whereby the mediating variable explains why the IV enhances the DV, illustrating how knowledge sharing and flow experience explain the enhancement of creativity through curiosity and sports commitment. On the other hand, the author uses "social media usage among fitness coaches may impede the occurrence of flow experiences, resulting in a non-significant relationship between flow experience and creativity" to explain why social media usage cannot moderate the relationship between flow experience and creativity. However, according to the data of this study, social media usage and flow experience seem to be positively correlated (since the author did not provide significance, this cannot be confirmed), hence this explanation is inconsistent with the study results, an alternative explanation is warranted.

---

## Round 0.2 · Minor Revisions

Dear Authors

Your revision has been reviewed by the experts in the field, and we recognize that the quality of the manuscript has been improved. However, there are two methodological concerns raised by the reviewers that should be addressed. Please highlight the point regarding the absence of a pilot test as a limitation of the study. Additionally, the methodological report for the Confirmatory Factor Analysis should be explicitly explored in the revision. Please submit a list of changes or a rebuttal against each point which is being raised when you submit the revised manuscript.

Best Regards

Yung-Sheng Chen, Ph.D.
Academic Editor

Reviewer 1 ·

Basic reporting

Improved version of manuscript.

Experimental design

Overall, you have improved the manuscript. However there are some issues which contradicted the present research practice.
1. No Pilot test was conducted: Conducting a formal pilot test remains methodologically superior, particularly because you adapted and combined several different instruments.
2. Limited information of CFA reported in the model: CFA is important in several ways, especially study on mediation effect. You only report GOF which is not sufficient. Please provide more indices eg. RMSEA, TLI, CFI, SRMR etc.

Validity of the findings

Please provide more fit indices

Reviewer 2 ·

Basic reporting

After the authors’ revisions, the rationale and necessity of the study have been sufficiently justified. The literature cited is representative and provides a solid theoretical foundation for the research. The manuscript is well-written, with appropriate structure, and the figures and tables are clearly and effectively organized.

Experimental design

The subject of the current manuscript, titled “Exploring the Impact of Curiosity and Sport Commitment on Creativity among Fitness Coaches: The Mediating Role of Knowledge-Sharing and Flow Experience,” is both intriguing and pertinent to the readership of this journal. This study tackles a significant and relevant issue concerning the interplay between curiosity, sport commitment, knowledge-sharing, flow experience and creativity. The introduction offers a comprehensive background on the current topic, linking fitness coaching with creativity; muitiple mediations and moderated mediations. The appropriate methodology employed further enhances the study's contribution by offering a comprehensive perspective on the mechanisms that underlie creativity.

Validity of the findings

The findings are supported by robust statistical evidence and extensive validity checks. The authors employed a comprehensive analytical strategy that includes convergent and discriminant validity assessments, common method bias tests, and bootstrapping procedures for mediation and moderation analyses. The reported R² values (e.g., 0.671 for creativity) suggest substantial explanatory power, and all hypothesized pathways were statistically supported with appropriate effect sizes and confidence intervals. The authors also addressed model fit using GOF indices and provided evidence for the reliability and internal consistency of all constructs. Overall, the results are statistically sound, conceptually coherent, and offer meaningful contributions to both theory and practice in sport psychology and fitness coaching domains.

---

## Round 0.3 · accepted · Accept

Dear Authors

Your revision has been reviewed by myself, and we recognize that the quality of the manuscript has been improved. Your submission is now endorsed for acceptance of publication in PeerJ.

In the revision, I strongly suggest to report the result of the pilot test in Results section.

Thank you for submitting your article to PeerJ. I would like to express my gratitude for your contributions and efforts to the scientific community. I look forward to receiving your research and review articles in the future.

Best Regards

Yung-Sheng Chen, Ph.D.
Academic Editor